# Quantum fluids of light in all-optical scatterer lattices

S. Alyatkin [1,2], H. Sigurdsson [1,2,3], A. Askitopoulos[1,2], J. D. Töpfer [1,2] & P. G. Lagoudakis[1,2,3 ✉]

One of the recently established paradigms in condensed matter physics is examining a system's behaviour in artificial potentials, giving insight into phenomena of quantum fluids in hard-to-reach settings. A prominent example is the matter-wave scatterer lattice, where high energy matter waves undergo transmission and reflection through narrow width barriers leading to stringent phase matching conditions with lattice band formation. In contrast to evanescently coupled lattice sites, the realisation of a scatterer lattice for macroscopic matter-wave fluids has remained elusive. Here, we implement a system of exciton-polariton condensates in a non-Hermitian Lieb lattice of scatterer potentials. By fine tuning the lattice parameters, we reveal a nonequilibrium phase transition between distinct regimes of polariton condensation: a scatterer lattice of gain guided polaritons condensing on the lattice potential maxima, and trapped polaritons condensing in the potential minima. Our results pave the way towards unexplored physics of non-Hermitian fluids in non-stationary mixtures of confined and freely expanding waves.

[1] Center for Photonics and Quantum Materials, Skolkovo Institute of Science and Technology, Moscow, Russia. [2] Laboratories for Hybrid Photonics, Skolkovo Institute of Science and Technology, Moscow, Russia. [3] School of Physics and Astronomy, University of Southampton, Southampton, UK. ✉email: p.lagoudakis@skoltech.ru

Artificial lattices are highly attractive for obtaining insight into properties of crystal structures in the solid-state, and for creating patterned structures not found in nature. They can be used to investigate spin frustration in triangular geometries[1], massless Dirac fermions in honeycomb structures[2], and strongly correlated states in dispersionless flat bands[3]. Conventionally, the physical properties of lattices are investigated through an appropriate choice of confined single-particle states (Wannier functions), such as electrons bound to their atoms, ultracold atoms in optical traps[4], or index-guided electromagnetic waves in photonic crystals[5]. The evanescent transfer of energy between lattices sites is then captured within the tight-binding approximation to explain crystal band formation, the centrepiece of solid-state physics[6].

However, in contrast to lattices of tightly confined states there exists the inverse case of coherent matter-wave scattering in the diffractive regime of lattices made up of repulsive potentials much smaller in size than the lattice spacing. There, waves with wavelength smaller than the lattice constant experience strong reflection and diffraction from the lattice (e.g., Bragg's law). Since the early work of Kronig and Penney[7], an ordered arrangement of spherically symmetric scatterers is known to give rise to bands and bandgaps. The study of elastic scattering of incoming and outgoing states on a static object is at the heart of mechanical, electromagnetic-, Schrödinger-, and sound-wave scattering theory[8]. However, the realisation of lattices of scatterers acting on matter waves with coherence length exceeding the scatterer spacing is much harder than the implementation of lattices of confined (evanescently coupled) wavefunctions, and the connection between the two regimes remains mostly unexplored.

Semiconductor microcavities in the strong coupling regime are especially appealing for engineering artificial lattices as they host matter-wave modes known as exciton-polaritons (from here on polaritons)[9]. These modes possess large nonlinearities, picosecond scale response times, and permit easy optical write-in and read-out of information. Polaritons can undergo power-driven nonequilibrium Bose−Einstein condensation, making them favourable candidates to study low threshold room-temperature lasing[10], optical many-body hydrodynamic phenomena[11,12], topological phases[13,14], and implementation of optical information processing[15]. Moreover, strong interparticle interactions result in repulsion of condensate polaritons from a background of uncondensed particles (i.e., photoexcited exciton reservoir), co-localised with the pumped area[16,17]. This enables all-optical design of non-Hermitian (i.e., optical gain and blueshift) potential landscapes for polaritons[18–20], in analogy to dipole optical traps for cold atoms[21] or photorefractive crystals[22]. Indeed, with non-Hermitian potentials the wavefunction norm is no longer conserved and, in conjunction with stabilising condensate nonlinearities, synchronisation can spontaneously appear amplifying the matter-wave similarly to phase-locked laser arrays[23,24].

Here, we demonstrate a non-resonantly optically imprinted repulsive (scatterer) lattice (see Fig. 1a, d) wherein scattered high energy polariton waves, emitted from the pump areas, result in robust interference patterns due to their ability to dynamically adjust their phase in order to condense into the highest gain Bloch state. Our scatterer lattice is chosen to have the edge-centred square (Lieb) arrangement, a configuration not found usually in nature, which offers comparison against the conventional tight-binding Lieb lattice (see Fig. 1c, f) which we also all-optically engineer. We perform full momentum-energy space tomography to unveil the engineered lattice band structures and their reshaping by altering the lattice parameters. We observe a gradual nonequilibrium phase transition from the scatterer lattice of ballistically expanding polariton waves (Fig. 1a, d) to a tight-binding lattice of trapped condensates (Fig. 1b, e) bridged by an unstable regime of multimodal condensation due to gain competition. Moreover, underscoring the flexibility of our optical approach, we provide direct observation of dispersionless P-flatband condensation achieved by using an excitation profile forming an "inverse" Lieb lattice (Fig. 1c, f) in the same spirit as the vacancy lattice created in electronic systems[25,26] or optical lattices of cold atoms[27,28]. We point out that the majority of our findings are not strongly dependent on the choice of Lieb lattice arrangement and can be extended to other types such as square, honeycomb, and triangle lattices.

## Results

**Scatterer Lieb lattice of polariton condensates.** A strain compensated $2\lambda$ GaAs-based planar microcavity with embedded three pairs of $In_{0.08}Ga_{0.92}As$ quantum wells[29] and an exciton-photon detuning of $-4$ meV is held at $\approx 4$ K in a closed-cycle helium cryostat. The non-resonant excitation (single-mode laser tuned at 1.5578 eV) is amplitude modulated at a frequency of 5 kHz (duty cycle 1%) with an acousto-optical modulator to avoid sample heating. A desired pump profile is shaped by a computer-controlled reflective phase-only spatial light modulator[30] and projected onto the sample through a microscope objective (NA = 0.42). We collect the real space polariton photoluminescence (PL), directly proportional to the condensate density, in transmission geometry and spectrally filter it from the excitation laser. We denote the horizontal and vertical momentum space coordinates as $\mathbf{k} = (k_x, k_y)$, corresponding to the spatial frequencies along horizontal ($x$-axis) and vertical ($y$-axis) real space coordinates, respectively.

As mentioned above, the flexibility in tuning the lattice properties through optical excitation structuring makes a system of microcavity polaritons appealing to access various lattice physics in a recyclable setting. The height of the potential landscape is determined by the excitation intensity and, in this study, constructed by an arrangement of Gaussian-shaped pump spots. On one hand, in the scatterer lattice where the pump spots have large separation distances in multiples of the polariton wavelength, polaritons condense on the maxima of the pumped potentials (see Fig. 1a, d) characterised by gain guided bright centres and ballistically expanding envelopes[19,20,30,31]. On the other hand, for closely spaced pump spots, they can condense in the minima of the potential landscape (see Fig. 1b, c, e, f) becoming optically trapped[32–34] partly because of their strong interactions that help them relax in energy. In the former case, the system shares analogies with antiguided laser arrays[24], whereas in the latter case with two-dimensional (2D) electron transport[2].

In Fig. 2a we show the real space PL of an optically pumped scatterer lattice with 96 polariton condensates at threshold power ($P = P_{thr}$) arranged into a Lieb geometry (corresponding to Fig. 1a, d). Each condensate is strongly gain guided with a bright centre co-localised with its respective pump spot. The size of the Gaussian pump spots sustaining the condensates is $\approx 2$ μm full width at half maximum and the lattice constant is set to $D = 20.3$ μm. As schematically shown in the inset of Fig. 2a, the Lieb lattice is composed of three square sublattices denoted with the letters A, B, and C. Being close to threshold, the spatial coherence of each condensate does not extend outside its respective pump spot[35] making them isolated (uncoupled) objects. Figure 2b, e shows the real and momentum space polariton PL above threshold ($P = 1.2P_{thr}$), while Fig. 2c, d shows energy −momentum space PL along $k_x = 0$ and $k_x = 2\pi/D$, respectively. By driving the system above threshold, we increase both the coherence and particle outflow from each condensate resulting in stronger coupling between the radiating condensates and the appearance of interference fringes (signature of synchronisation)

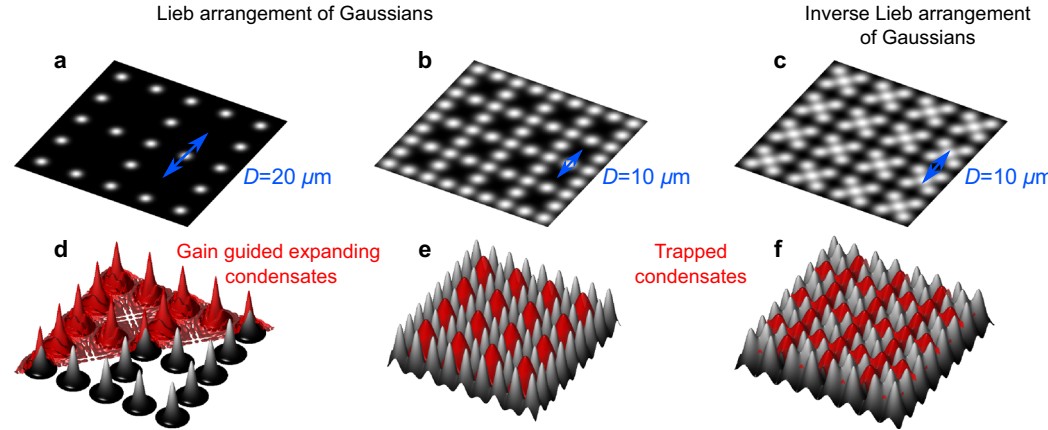

**Fig. 1 Schematic of the optical excitation pattern and resulting polariton condensates. a, b** Excitation intensity profile composed of Gaussian pump spots arranged in a Lieb pattern for two different lattice constants $D$ and **c** an inverse Lieb pattern with the potential minima (dark areas) forming a conventional Lieb lattice. **d–f** Corresponding black-white surface plots of the pump induced potential landscapes with the polariton condensate density overlaid as red envelope. In **d** polariton condensation occurs on the potential maxima (gain guided condensates) resulting in highly energetic (ballistic) condensate waves whereas in **e**, **f** condensation takes place in the potential minima between pump spots.

**Fig. 2 Scatterer Lieb lattice of polariton condensates. a** Measured PL from a scatterer lattice of polariton condensates with a lattice constant set to $D = 20.3 \, \mu m$ excited at $P = P_{thr}$ and **b** at $P = 1.2P_{thr}$. Inset in **b** shows the calculated Bloch state density for comparison. The inset in **a** shows sublattices denoted with letters A, B, C forming the lattice. **c, d** Energy-resolved momentum space polariton PL at $P = 1.2P_{thr}$ for $k_x = 0$ and $k_x = 2\pi/D$ values respectively, marked on **e** with vertical lines and yellow letters "c" and "d". **e** Energy integrated momentum space polariton PL at $P = 1.2P_{thr}$, and **f–h** energy-resolved "slices" of momentum space at energies 1.6, 1.3, and 1 meV with respect to the bottom of the lower polariton branch (dispersion). **i** Calculated lattice bands with the red circle corresponding to the Bloch state in the inset in (**b**). **j–l** Numerically calculated "slices" of polariton momentum space PL from Monte-Carlo simulations on the 2D dissipative Schrödinger equation.

as well as distinct Bragg diffraction peaks in momentum space. Specifically, we observe the condensates populating an excited Bloch state characterised by a constructive interference peak in the centre of each lattice cell, also visible from the calculated Bloch state shown in the inset in Fig. 2b. By scanning the Fourier space PL (collected in reflection geometry) image across a slit of a spectrometer with a motorised mount, we reconstruct energy-resolved "slices" (i.e., isoenergy planes) of momentum space PL at three specific energies, shown in Fig. 2f–h. The applied energy tomography reveals the full picture of complex band formation (see Supplementary Movie 1). We also calculate the band structure using two different modelling methods: Bloch's theorem, and Monte-Carlo sampling of the 2D dissipative polariton Schrödinger equation (see Supplementary Notes 2 and 3). The Bloch analysis, as shown in Fig. 2i, reveals a zoo of overlapping bands of distinct shapes, yet the polaritons, being so interactive, are still easily able to relax into a definitive Bloch state corresponding to the optimum gain (marked with a red circle and plotted in Fig. 2b inset), in a similar spirit to coupled laser systems. The numerical Monte-Carlo sampling of the dominant Fourier components of the scatterer lattice shown in Fig. 2j–l gives good agreement with the experiment. We note that the illuminated, clearly formed, bands in Fig. 2c, d show that polaritons in the repulsive scatterer Lieb lattice indeed experience crystal scattering within their coherence time. We also observed band-structure formation for the square scatterer lattice of $5 \times 5$ polariton condensates with a similar lattice constant (see Supplementary Note 1).

**Lattice constant dependence**. Next, we decrease the lattice constant from $D = 20\,\mu m$ to $D = 10.3\,\mu m$ and characterise the change in the polariton system behaviour. Corresponding images

with experimentally measured real space polariton PL above condensation threshold are depicted in Fig. 3a–e. As shown in Fig. 3a, for $D = 20\,\mu m$, most of the PL intensity comes from the pumped areas with clear interference fringes in time-averaged measurements as a consequence of the scattered polariton waves leading to robust synchronisation between the condensates[30,31]. Multiple weakly populated energy branches collapse into one dominant mode as the pump power exceeds the threshold value, as clearly seen in Fig. 3f. Decreasing the lattice constant to $D = 16.9\,\mu m$ results in substantial growth of PL intensity inside each lattice cell (i.e., where a Gaussian pump spot is absent) with simultaneous decreased PL at the pumps positions (see Fig. 3b). Driving the system above threshold leads to dual-mode condensation as confirmed by the measured spectrum power scan shown in Fig. 3g. Decreasing the lattice constant to $D = 15.2\,\mu m$ again dramatically modifies the polariton PL pattern. As shown in Fig. 3c, polaritons are repelled even stronger outside the pumped areas leading to complex PL distribution in real space. The condensate is here fractured into multiple energy modes above threshold (see Fig. 3h), with a similar gain. Finally, decreasing to even smaller lattice constants of $D = 12.3\,\mu m$ and $D = 10.3\,\mu m$ results in the formation of trapped condensates[33] as shown in Fig. 3d, e, characterised by a dominant single energy mode above threshold (see Fig. 3i, j). This regime is schematically depicted in Fig. 1b, e.

The most intriguing physics lies in the intermediate regime where multimodal condensation takes place between the regimes of gain-guided ballistically coupled condensates (Fig. 3a) and trapped condensates (Fig. 3d, e). In order to distinguish these regimes and identify the transition between them we introduce a contrast parameter $S = (I_{spots} - I_{trapped})/(I_{spots} + I_{trapped})$ describing the

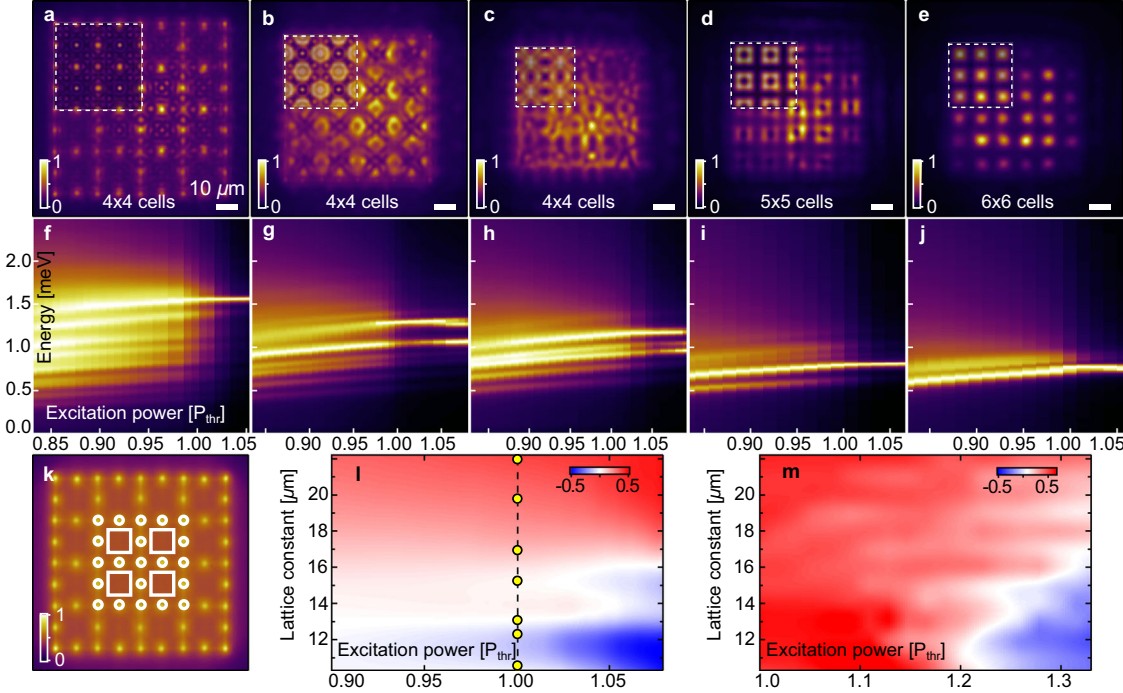

**Fig. 3 Optical lattices of polariton condensates pumped with Lieb geometry for different lattice constants ($D = 20.0, 16.9, 15.2, 12.3, 10.3\,\mu m$).**
**a–e** Shows corresponding real space polariton PL above condensation threshold and **f–j** corresponding spectra as functions of pump power (energy is scaled with respect to the bottom of lower polariton branch). Semi-transparent insets in **a–c**, **e** marked with white dashed squares show results of time-averaged numerical simulations of the condensate dynamics using the generalised Gross–Pitaevskii equation. Inset in **d** is calculated using Bloch's theorem. White circles and squares overlaid with real space PL below threshold for $D = 20\,\mu m$ in (**k**) denote spatial integration areas on top of and outside the pump spots used to extract the contrast between gain guided ($I_{spots}$) and trapped ($I_{trapped}$) polariton PL. This contrast is plotted as normalised heatmaps $S = (I_{spots} - I_{trapped})/(I_{spots} + I_{trapped})$ in (**l**, **m**) as a function of lattice constant and pump power from experiment and simulations, respectively, revealing a gradual transition between the two regimes. Scale bar in (**a**) applies also to (**b–e**, **k**). Yellow circles in **l** denote the lattice constants realised experimentally.

normalised difference between the average polariton PL at the pumping spots $I_{spots}$ (white circles in Fig. 3k) and outside the spots $I_{trapped}$ (white squares in Fig. 3k). Here, "average" refers to area-integrated PL divided by the integration area. The side length of the white squares is chosen as $L = 0.54D$ (the precise value does not affect the findings) whereas the diameter $d = 3.6\,\mu m$ of the white circles is fixed for all lattice constants.

In the case of well-separated pump spots (i.e., $D > 17\,\mu m$) we observe values $S > 0$ in Fig. 3l across all pump powers corresponding to high-energy gain guided polaritons that scatter across the lattice when emitted from their pump spots. In the opposite case of small separation distances between the pumps (i.e., $D < 12.5\,\mu m$) we observe $S < 0$ corresponding to polaritons becoming trapped in the lattice potential minima with dominant PL intensity coming from the white squares in Fig. 3k. Here, high energy scattering of polaritons across multiple lattice cells is reduced, replaced with the evanescent transfer of energy (i.e., tunnelling). Therefore, at some lattice constant (where $S \approx 0$) the polariton system undergoes a none-quilibrium phase transition between these different condensation regimes. During this transition, competing lattice modes fight over the gain which leads to a "frustrated" condensation pattern, like shown in Fig. 3b, c. This transition is unique to polariton systems since it is the excitonic part of polaritons which allows them to interact with each other and the uncondensed exciton reservoir, which facilitates energy relaxation[16]. In contrast, purely photonic systems would generally lase in their pumped gain region. The gradual onset of trapped polaritons as the lattice constant reduces and pump power increases (i.e., the background uncondensed reservoir increases) can therefore be attributed to modes between the pump spots moving into resonance with the lattice gain-bandwidth due to enhanced energy relaxation of polaritons[36]. We point out that our all-optical experiment and extraction of $S$ for different lattice constants and pump powers is not possible to replicate with lithographically written photonic periodic structures where the pumped gain region would correspond to the lattice potential minima[13]. In our experiment, the pump gain follows the lattice potential maxima. We qualitatively reproduce our findings through numerical mean-field simulations using the generalised Gross−Pitaevskii equation[37] including an energy relaxation mechanism[36] (see Supplementary Note 4 for details), shown in the insets of Fig. 3a–c, e, m. The state in the inset in Fig. 3d was calculated using Bloch's theorem and did not appear in a stable form in Gross−Pitaevskii simulations for our chosen (fixed) set of simulation parameters. Some discrepancy can be observed between simulation and experiment in Fig. 3l, m at low powers and small lattice constants where the simulation overestimates the gain guided polaritons. This discrepancy could be reduced by applying a stochastic treatment (e.g., Wiener noise) to the condensate equations of motion which would smear out the simulated condensate PL at low powers close to threshold, or by including exciton diffusion in the model.

**Flatband polariton condensation**. To further demonstrate the versatility of our all-optical approach we move away from the scatterer Lieb lattice and design now the conventional Lieb lattice (see Fig. 1c, f) which has been studied vigorously in various systems of electrons[25,26], photons[38–40], cold atoms[27,41], plasmon polaritons[42], and exciton-polaritons[43–47]. It hosts an excess of phenomena including topological phases, dispersionless flat bands, and Dirac points making it a popular testbed in solid-state physics. In Fig. 4 we realise the conventional Lieb lattice by arranging the Gaussian excitation spots in an inverse Lieb lattice (Fig. 4a, e). Just as in Fig. 3d, e, here the polariton condensates are designed to be trapped in the lattice potential minima, interacting

via tunnelling. Figure 4b, c shows the real- and momentum space polariton PL above threshold, respectively, in a $D = 13\,\mu m$ lattice revealing condensation into the S-band ground state in the Γ point at $k = 0$. By adjusting the lattice parameters, the condensate can be forced to populate a different Bloch state. Figure 4e–g shows the same measurements but for $D = 16\,\mu m$ where we now observe condensation into P-orbitals at sublattices A and C, whereas at sublattice B destructive interference occurs. This state belongs to a dispersionless P-flatband[43,44] and constitutes the first evidence of polariton condensation into optically (non-Hermitian) generated flatband states. Figure 4d, h shows corresponding calculated lattice states using Bloch's theorem. The state plotted in Fig. 4h corresponds to the band denoted by the red circle in Fig. 4i where we show the calculated P-bands along the reduced Brillouin zone edge. We additionally calculate the dispersion of the lattice from Monte-Carlo sampling of the dissipative Schrödinger equation and plot the results in Fig. 4j along $k_x = 2\pi/D$. The results reveal bright bands where polaritons decay slowest with the P-flatband clearly visible (marked with yellow $k$) and the lowest energy P-band (marked with yellow $k$') which only appears flat along the Γ-X direction.

Experimentally implemented energy tomography measurements allow us to extract PL belonging to isoenergy planes in reciprocal space which further verifies that the condensate is populating the P-flatband. Figure 4k, l shows the calculated and measured PL from the P-flatband isoenergy plane, respectively, which is marked with the yellow-coloured letter $k$ in Fig. 4j. Indeed, the similarity between Fig. 4l and the energy-integrated PL in Fig. 4g confirms that condensate is dominantly populating a single energy state with a similar intensity pattern obtained from calculation (see Supplementary Note 1 for more details). It should be noted that the calculated PL (Fig. 4k) shows some finer details in momentum space which would average out in time-integrated measurements due to non-ideal effects such as noise, cavity disorder, and pump fluctuations. However, the qualitative structure is the same as in experiment with most of the PL belonging to the $M$ symmetry points. The fact that the PL is strongest at the $M$ points can be understood from the slight curvature in the P-flatband around these points (see Fig. 4i) which facilitates the relaxation of polaritons into the band minima.

We now address the question on why condensation occurs into the P-flatband as shown in Fig. 4f, g. As mentioned in the previous paragraph, the P-flatband in our optical lattice is not perfectly flat and is characterised by a small dispersion (curvature) around the $M$ point which is only $\approx 5\,\mu eV$ different in energy from both the Γ or the $X$ points (see Fig. 4i). This small band curvature can trigger condensation into the band minima and is attributed to the finite potential depth of the lattice sites (taken here to be 2 meV in calculation) which leads to deviation from the perfectly dispersionless bands predicted by the tight-binding theorem[27]. The band curvature can be reduced by increasing the potential depth of the inverse Lieb lattice (see Supplementary Note 5) which can be achieved by adjusting the system properties (e.g., exciton-photon detuning or the exciton dipole moment by appropriate choice of the semiconductor material) such that stronger pumping (bigger blueshifting reservoir) is required to achieve condensation. Another feature of our inverse optical Lieb lattice is that it creates slightly elliptical confinement at sublattices A and C which splits the energies and linewidths of the $P_{x,y}$ orbitals. In other words, the overlap of $P_{x,y}$ orbitals into the pump (gain) region is different. For sublattice A (C) the $P_x (P_y)$ orbital overlaps more with the pump which creates a higher gain for the P-flatband state. This interpretation is supported through a non-Hermitian tight-binding theory (see Supplementary Note 6).

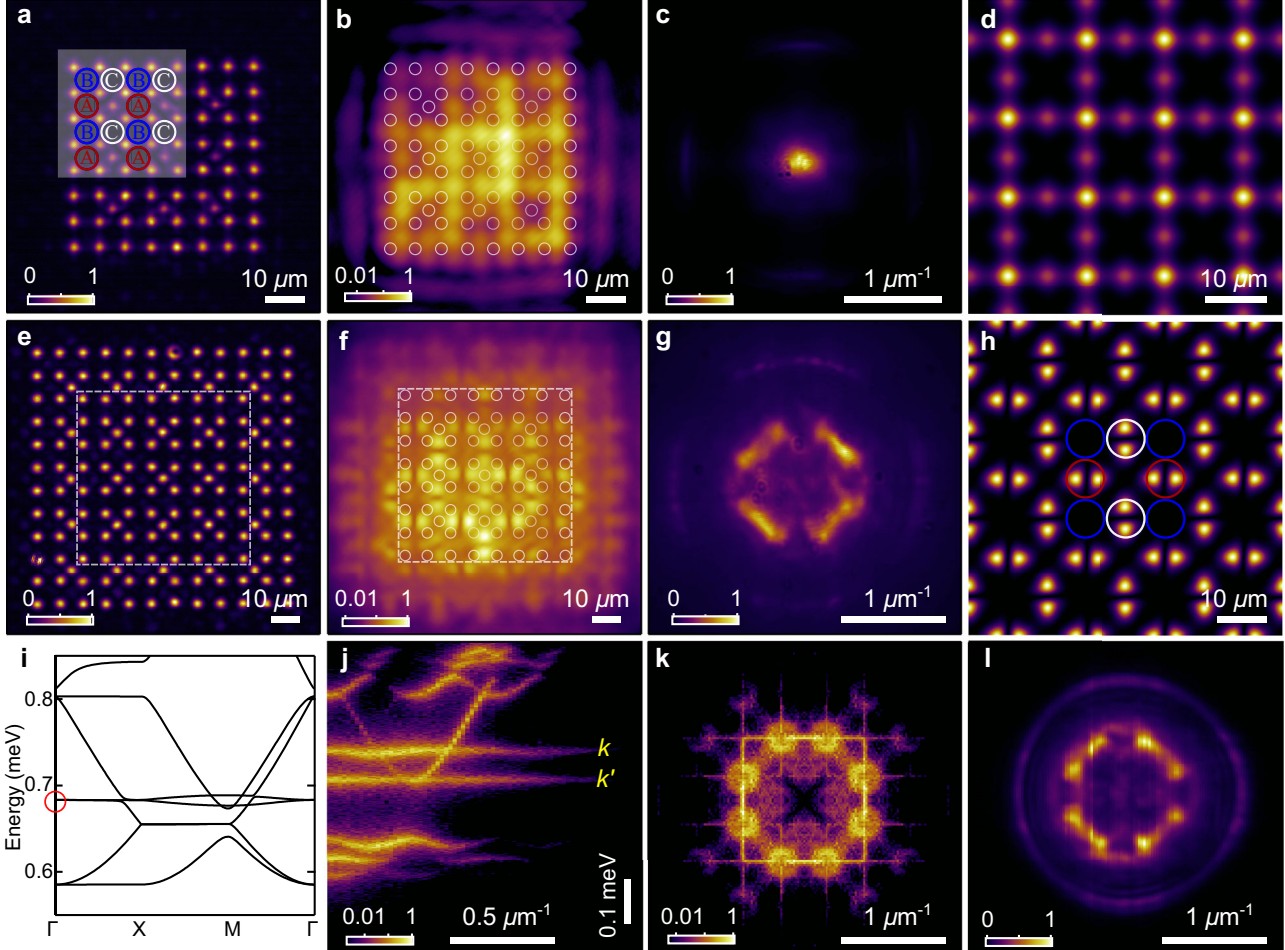

**Fig. 4 Optical inverse Lieb lattice of Gaussians with flatband condensation. a, e** Spatial pumping profile of Gaussian spots arranged in the inverse Lieb geometry with lattice constants $D = 13$ and 16 µm, respectively. In **a** the cube-centred spots are 40% weaker in intensity compared to the rest of the pump spots. In **e** all spots are equally intense. **b, f** Real and **c, g** momentum space polariton PL above condensation threshold corresponding to populated S-band ground state and P-flatband state, respectively. White circles in **b, f** denote the pumps positions. **d, h** Corresponding calculated real space wavefunction densities intensities from Bloch's theorem. Coloured circles in **a, h** denote the Lieb lattice unit cells. **i** Calculated P-bands from Bloch's theorem where the red circle marks the flatbands. **j** Calculated dispersion cross-section at $k_x = 2\pi/D$ from Monte-Carlo Schrödinger numerics. Both **i** and **j** use the pump profile shown in (**e**). **k, l** Calculated and measured PL from the isoenergy plane in momentum space corresponding to the P-flatband marked with the yellow letter k in (**j**). The band marked with the yellow letter k' is the lowest energy P-band which only appears flat between the Γ-X points in the Brillouin zone.

## Discussion

The demonstrated optically arranged system of exciton-polaritons presents a flexible platform to study fundamental proposals on non-Hermitian phenomena in artificial lattices including non-linear reshaping of the crystal bands when above condensation threshold. In Fig. 2 we have demonstrated the condensation of polaritons into excited Bloch states belonging to a lattice of scatterers (narrow waist repulsive potentials) shaped into a Lieb pattern. The observed crystal bands and agreement with theoretical models opens up a path to explore more intriguing effects of scattered matter-waves such as slow polaritons[48], guided polaritons[49], and solitonic modes[50]. In Fig. 3 we have revealed a gradual nonequilibrium phase transition from the scatterer Lieb lattice, characterised by strongly gain guided and energetic polariton condensates, to a square lattice of optically trapped condensates as a function of two easily tunable parameters in the experiment, the lattice constant and pump power. The transition regime is accompanied by multimode polariton lasing, which is a unique feature of polariton systems due to their strong interactions, that lead to unexpected condensation patterns in real space. Such multimodal behaviour of the condensate implies close gain

competition between gain-guided and trapped polaritons. Our observations, therefore, highlight the intriguing duality of polaritons as "lasers" (gain guiding) and "Bose−Einstein condensates" (thermalisation) with interesting perspectives as strongly non-linear objects in non-Hermitian optical lattices. As an example, the optical malleability of our matter-wave platform allows one to study in detail the gradual departure from a system of quantised energies to that of smoothly connected quasimomentum states by building the lattice up cell-by-cell.

Another possible perspective is to optically engineer the lattice to probe exotic band properties for polariton condensation. In Fig. 4 we have demonstrated a conventional (i.e., tightly bound waves) Lieb lattice, by packing pump spots into its inverse shape, with subsequent condensation into flat band states. Given the nonlinear nature of exciton-polaritons, condensation into adjustable flat band states could open a window to investigate strongly correlated states of matter and the effects of disorder against interactions[51]. The implemented energy tomography methods provide complete access to the polariton states in Fourier space and can be used to study the reshaping of the spectrum under arbitrary lattice deformation and nonlinear effects. We point out

that the current study is performed in the scalar polariton regime, but can be easily extended to include its spin degree of freedom by changing the pump polarisation. Furthermore, bands induced by our optically engineered landscape can be populated using resonant excitation permitting the study of the evolution of polariton matter waves in non-Hermitian optical lattices with any chosen crystal momentum and frequency. We believe that our work carries significant weight in the future design and investigation of polaritonic non-Hermitian (gain and loss) lattice physics in e.g., topological lasers[52,53], phase transitions in many body systems[54,55], non-reciprocal transport[56], and access to a multitude of gain-induced anomalies reported for diffractive metasurfaces[57].

## Data availability

Data supporting this study are openly available from the University of Southampton repository at https://doi.org/10.5258/SOTON/D1947.

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

## Acknowledgements

We acknowledge the support of the UK's Engineering and Physical Sciences Research Council (grant EP/M025330/1 on Hybrid Polaritonics). S.A. acknowledges the support of the Russian Science Foundation (RSF) grant No. 21-72-00088. H.S. and P.G.L acknowledge support by the European Union's Horizon 2020 programme, through a FET Open research and innovation action under the grant agreement No. 899141 (PoLLoC).

## Author contributions

P.G.L. led the research project. P.G.L., S.A., A.A., and J.D.T. designed the experiment. S.A., A.A., and J.D.T. carried out the experiment and analysed the data. H.S. developed the theoretical modelling and performed numerical simulations. All authors contributed to the writing of the paper.

## Competing interests

The authors declare no competing interests.
