## [Peer Review File · Nature Communications]

Reviewers' Comments:

Reviewer #1:

Remarks to the Author:

In "all-optical non-Hermitian point scatterer lattices for matter wave fluids", Alyatkin et al. investigate band structure formation in all-optical polariton lattices. Using the versatility of the system in terms of lattice geometry and lattice spacing, they explore two different regimes: (i) For large lattice spacing, the polaritons condense on the potential maxima and coupling between condensates occurs through ballistic expansion due to the repulsive interactions induced by the non-resonantly excited exciton reservoir (repulsive Dirac comb potential). (ii) For smaller lattice spacing, polariton condensation occurs in the potential minima and coupling between condensates is due to the evanescent coupling between sites (tight-binding approach). The authors perform the full tomography of the system (real space and reciprocal space) to characterize the lattices band structures and polariton condensation in the state of highest gain. The data are of high quality and the agreement with simulations is compelling.

Although I am able to appreciate the quality of the results, my main problem with this article is related to its organization and its story telling. The main message the authors want to convey is not clearly stated:

1- The title and the abstract exclusively mention "non-Hermitian point scatterer lattices", which seems to be the main focus of the article. However, one third of the work is dedicated to a totally different regime. This gives the impression that the article is composed of two disconnected parts that could be part of two distinct works. I find this confusing. As a matter of fact, the authors themselves have chosen a different title for the supplemental material ("Interaction induced point scatterer lattices and flat band condensation of exciton-polaritons"), which may have been an attempt to include the part of the work covered in Fig.4.

2- In the current form of the article, my understanding is that the main message is the implementation of all-optical polariton potentials, and the observation of band structures in these potentials. This may not be sufficient since the implementation of all-optical potentials for polaritons and band structure formation has already been reported in the following works: Askitopoulos, H. Ohadi, A. V. Kavokin, Z. Hatzopoulos, P. G. Savvidis, and P. G. Lagoudakis, "Polariton condensation in an optically induced two-dimensional potential", Phys. Rev. B 88, 041308(R) (2013).

H. Ohadi, Y. del Valle-Inclan Redondo, A. J. Ramsay, Z. Hatzopoulos, T. C. H. Liew, P. R. Eastham, P. G. Savvidis, and J. J. Baumberg, "Synchronization crossover of polariton condensates in weakly disordered lattices", Phys. Rev. B 97, 195109 (2018).

J. D. Töpfer, H. Sigurdsson, L. Pickup & P. G. Lagoudakis, "Time-delay polaritonics", Communications Physics 3, 2 (2020).

L. Pickup, H. Sigurdsson, J. Ruostekoski & P. G. Lagoudakis, "Synthetic band-structure engineering in polariton crystals with non-Hermitian topological phases", Nature Communications 11, 4431 (2020).

As a consequence, it would be beneficial to shift the focus of article and formulate it in a way that more emphasis is placed on the physical effects at play (laser synchronization and condensation dynamics into given modes, flat band physics...) rather than on the implementation itself.

Below are some other comments:

1- Fig.4 caption needs work. Some of the referencing is inaccurate and I cannot see the "white circles in (c) and (h)" for example.

2- Fig.2 and Fig.3 show k vs E plots of the band structures. Why are k vs E plots not shown in Fig.4? Can we see the measured band structure?

3- The calculated band structures in Fig.4k and Fig. S5c shows some residual band curvature. Can the authors comment on that? Is there potentially a way to improve band flatness?

4- Fig.4n and Fig.4o show condensation into two different P-bands (labelled l and m). Is this achieved by varying the laser excitation power? The lattice configuration? Is there a way to controllably decide in which mode condensation is going to occur?

5- The authors comment on the inset of Fig.4o as being potential evidence of "a strongly interacting particle phase, inherent to a flat band state". There is not enough evidence to support this claim and the system is definitely not in the strongly interaction regime. That statement should be removed.

6- Towards the end of the paper comes a discussion of "two faulty spots in the excitation pattern in Fig.4f". That statement is misplaced and should appear either earlier in the article (when

describing the technical implementation) or in the supplemental material.

Reviewer #2:

Remarks to the Author:

Lieb lattices are well known to give rise to flatbands, especially in the case of decorated square lattices. Flatband topological states have been reported in many photonic systems, including very convenient arrangements like arrays of waveguides (ref [36]) or plasmonic scatterers (Phys. Rev. B 93, 075126 (2016)). The flatbands result from interference of the scattered coherent light with the periodicity of the scattering lattice. In this manuscript the authors offer a very small variation on the already well established literature on the subject. As a source of coherent light they use microcavity polaritons in their lasing state, and they build the decorated square lattice by locally changing the refractive index of a planar microcavity with an external laser. It is not clear, throughout the paper, what would be advantage of using polaritons operated at 4 K with respect to room temperature systems like laser arrays or waveguide arrays. On the contrary, low temperature operation looks like a stepback. While technically correct, this lack of novelty of the present work makes it more suitable for a more specialistic journal.

Reviewer #1 (Remarks to the Author):

In "all-optical non-Hermitian point scatterer lattices for matter wave fluids", Alyatkin et al. investigate band structure formation in all-optical polariton lattices. Using the versatility of the system in terms of lattice geometry and lattice spacing, they explore two different regimes: (i) For large lattice spacing, the polaritons condense on the potential maxima and coupling between condensates occurs through ballistic expansion due to the repulsive interactions induced by the non-resonantly excited exciton reservoir (repulsive Dirac comb potential). (ii) For smaller lattice spacing, polariton condensation occurs in the potential minima and coupling between condensates is due to the evanescent coupling between sites (tight-binding approach). The authors perform the full tomography of the system (real space and reciprocal space) to characterize the lattices band structures and polariton condensation in the state of highest gain. The data are of high quality and the agreement with simulations is compelling.

Response: We thank reviewer for carefully reading of the manuscript and high evaluation of our results.

Reviewer 1-1: Although I am able to appreciate the quality of the results, my main problem with this article is related to its organization and its story telling. The main message the authors want to convey is not clearly stated:

1- The title and the abstract exclusively mention "non-Hermitian point scatterer lattices", which seems to be the main focus of the article. However, one third of the work is dedicated to a totally different regime. This gives the impression that the article is composed of two disconnected parts that could be part of two distinct works. I find this confusing. As a matter of fact, the authors themselves have chosen a different title for the supplemental material ("Interaction induced point scatterer lattices and flat band condensation of exciton-polaritons"), which may have been an attempt to include the part of the work covered in Fig. 4.

Response: Indeed, in this work we aimed to show all advantages of the implemented all-optical approach by demonstrating two distinct regimes in one system as pointed out by the Reviewer. The former diffractive regime concerns scatterer type lattices, where the excited polariton mean-wavelength is smaller than the lattice constant, which has not been explored properly for quantum fluids. The latter regime concerns deeply confined states in a tight-binding lattice which are more commonly studied (e.g., ultracold atoms). We fully agree with the Reviewer that the flow of our study possessed a discontinuity between the regimes, and the connection between the two was not clear.

To strengthen the novelty and to clarify the goal of our work, we have now added new experimental measurements (see new Fig. 3) that bring these two distinct parts of our original manuscript closer together and highlight a previously unexplored intermediate (multimodal condensate) lattice regime. We gradually tune the lattice constant D of the scatterer Lieb structure in a controllable manner, from one regime to the other, and resolve the polariton photoluminescence (PL) in real space and energy resolved momentum space as a function of

pump power. We observed in an intermediate range of lattice constants ($D = 13 \leftrightarrow 16 \mu\text{m}$) the pump-repelled polaritons occupy several competing energy states above condensation threshold (i.e., multimodal lasing). This behaviour appears between the single-mode scatterer regime, inherent to the case of well separated pump spots that excite ballistically expanding condensates [$D = 20 \mu\text{m}$, see Figs. 3(a,f)], and the regime of tightly-bound (trapped) condensates [$D = 10.3 \mu\text{m}$, see Figs. 3(e,j)]. This new intermediate regime evidences a form of “gain frustration” where polaritons cannot decide whether to condense on top of their pump spots (i.e., gain guiding effect) or into the potential minima between the spots (i.e., energy relaxation effect).

In this new multimodal regime, the total PL intensity I_{spots} on top of the pump spots [see white circles in Fig. 3(k)] is very close to the total PL intensity I_{inside} within the inner area of the lattice cells [see white squares in Fig. 3(k)]. In order to get a clear figure of merit and to reveal the transition between the distinct regimes of our study we introduced a contrast parameter S which reads: $S = (I_{spots} - I_{inside}) / (I_{spots} + I_{inside})$. The extracted S value, as a function of lattice constant and pump power, is shown in Fig. 3(l) of the revised manuscript. From this figure one can see a clear transition area (white coloured) between the regime of polariton condensation on top of the pump spots (red coloured) and the regime of localised condensates trapped in between pump spots (blue coloured). We apply a generalised Gross-Pitaevskii model, describing the condensate dynamics with energy relaxation, and achieve good agreement with our experimental observations. We note that the observation of such a transition regime is practically not possible in photonic micropillars structures, which have been the basic platform for polariton lattices so far.

In order to clarify the main message and underline the novelty of our work we have incorporated the new results into the revised manuscript and changed its title to “Quantum fluids of light in all-optical scatterer lattices”. We have also corrected the title of the Supplementary Material (SM), and heavily rewritten our manuscript to increase the coherence of our message as suggested by Reviewer 1. To our knowledge, such a connected study for polaritonic lattices has not been presented in literature and, in-fact, we cannot find evidence that any other quantum fluid system has demonstrated the ability to continuously tune the lattice constant and the central frequency of the system gain bandwidth from one regime to the other.

Moreover, in the modified introduction we now stress that the choice of a Lieb lattice is not important to the message of the paper. We could have also studied a honeycomb lattice that possesses a wealth of physical phenomena and the transition from ballistic to trapped condensates by changing the lattice constant. We hope these additional efforts have made our message clearer.

Reviewer 1-2: In the current form of the article, my understanding is that the main message is the implementation of all-optical polariton potentials, and the observation of band structures in these potentials. This may not be sufficient since the implementation of all-optical potentials for polaritons and band structure formation has already been reported in

the following works: Askitopoulos, H. Ohadi, A. V. Kavokin, Z. Hatzopoulos, P. G. Savvidis, and P. G. Lagoudakis, "Polariton condensation in an optically induced two-dimensional potential", Phys. Rev. B **88**, 041308(R) (2013). H. Ohadi, Y. del Valle-Inclan Redondo, A. J. Ramsay, Z. Hatzopoulos, T. C. H. Liew, P. R. Eastham, P. G. Savvidis, and J. J. Baumberg, "Synchronization crossover of polariton condensates in weakly disordered lattices", Phys. Rev. B **97**, 195109 (2018). J. D. Töpfer, H. Sigurdsson, L. Pickup & P. G. Lagoudakis, "Time-delay polaritonics", Communications Physics **3**, 2 (2020). L. Pickup, H. Sigurdsson, J. Ruostekoski & P. G. Lagoudakis, "Synthetic band-structure engineering in polariton crystals with non-Hermitian topological phases", Nature Communications **11**, 4431 (2020). As a consequence, it would be beneficial to shift the focus of article and formulate it in a way that more emphasis is placed on the physical effects at play (laser synchronization and condensation dynamics into given modes, flat band physics...) rather than on the implementation itself.

Response: We fully agree with points raised by the Reviewer and have now modified the focus of our work such that it presents a comprehensive study on the new physics appearing in all-optically engineered two-dimensional lattices for polaritons across two extremely different regimes of scattered/refracted polariton waves and trapped/bound polariton states. Moreover, by gradually adjusting the lattice geometry and analysing its spectral and spatial PL behavior in the nonlinear lasing regime (above condensation threshold) we are able to unveil an intermediate multimodal regime (see our reply to question 1) which, to our knowledge, has not been reported in polariton lattices of any form.

We note that our study differs significantly from published earlier works mentioned by Reviewer 1 and below we explain why. The work by A. Askitopoulos et al., [Phys. Rev. B **88**, 041308(R) (2013)] concerns just a single optical trap for polariton condensation. The work by Ohadi et al. [Phys. Rev. B **97**, 195109 (2018)] covers synchronization for localized polariton condensates in a square lattice of optical traps. The work by J.D. Töpfer et al. [Commun. Phys. **3**, 2 (2020)] is focused on the dynamical coupling mechanism between only two ballistic polariton condensates. Finally, the paper by L. Pickup et al. [Nat. Commun. **11**, 4431 (2020)] deals with one-dimensional chains of ballistic condensates with focus on optically induced defect states. Our work is the first to show evidence of polariton band formation in two-dimensional all-optical lattices (across two very different regimes) but, more importantly, it studies an unexplored path linking the nonlinear lattice dynamics of waves experiencing scattering/refraction with that of deeply confined states, with an unreported dynamical intermediate phase where condensation (lasing) not only forms into multiple distinct momentum-space (Bragg) peaks but also across multiple energies.

Additionally, unlike all above-listed papers, our work uses the Lieb lattice which possesses fundamentally intriguing properties such as Dirac cones and dispersionless bands. As we mentioned in our previous answer to question 1 of the Reviewer, the choice of a Lieb lattice is not crucial to the main message of the paper. None-the-less, by choosing a Lieb lattice platform we are able to underscore the advantage of our all-optical method by constructing an inverse optical Lieb lattice (see Fig.4). This complements previous systematic studies on Lieb lattices in the tight-binding approximation across various platforms such as lithographically written photonic structures [Tang et al., Nanophotonics **9**, 1161 (2020)]. We

therefore strongly believe that our work will trigger widespread interest in the scientific field and stimulate research groups to pursue studies of non-Hermitian many-body physics in optically generated polariton lattices.

As we mentioned above, we have now heavily rewritten our manuscript to reflect enhanced understanding on the polariton system properties, its potential impact, and uniqueness. We feel it is impractical to point out all changes in the manuscript and therefore we will summarise the main ones here:

- 1) The change of manuscript title to: “Quantum fluids of light in all-optical scatterer lattices”
- 2) A modified introduction and Fig. 1 which more clearly introduces the idea of the two different regimes of gain-guided ballistic polariton condensates and trapped polariton condensates depending on the lattice constant (as compared to the mean condensate wavelength). Importantly, for the former we have introduced the notion of a “*diffractive regime*”. Physically, this means that the ballistic polariton condensate has a wavelength λ smaller than the pump spot spacing (and thus less than the lattice constant) making the matter wave fluid susceptible to reflection and refraction (especially if $\lambda/2 < D$) which is evidenced from the clear Bragg peaks in the momentum space PL of Fig. 2c. We point Reviewer 1 to the following modified sentence:

“However, in contrast to lattices of tightly confined states there exists the inverse case of coherent matter-wave scattering in the diffractive regime of lattices made up of repulsive potentials much smaller in size than the lattice spacing. There, waves with wavelength smaller than the lattice constant experience strong reflection and refraction from the lattice (e.g., Bragg's law).”

Additionally, in the introduction, we have clarified that the choice of the Lieb lattice arrangement is not the central part of our study's message. However, we feel that the results presented in Fig. 4 underscore the flexibility of our approach and nicely closes the storyline of the study. We point Reviewer 1 to the following modified sentence:

“We point out that the majority of our findings are not strongly dependent on the choice of Lieb lattice arrangement and can be extended to other types such as square, honeycomb, and triangle lattices.”

- 3) We have replaced previous Fig. 2 with previous Fig. 3 and introduced a new Fig. 3 with experimental and theoretical data elucidating on the transition from the gain-guided ballistic condensate regime to the trapped condensate regime. The text surrounding Fig. 3 is heavily modified from our previous manuscript to properly integrate the new findings into the study.

- 4) We now go beyond our linear models and have added two-dimensional generalised Gross-Pitaevskii equation simulations, describing the polariton condensate dynamics above threshold, which show good agreement between theory and experiment in Fig. 3.
- 5) We have made some modifications to Fig. 4 according to Reviewer 1 comments (see below) and now—importantly—explain the finite band curvature in the P-flatbands. We point the Reviewer to the following text:

“As mentioned in the previous paragraph, the P-flatband in our optical lattice is not perfectly flat and is characterised by a small dispersion (curvature) around the M point which is only $\approx 5 \mu\text{eV}$ different in energy from both the Γ or the X points (see Fig. 4i). This small band curvature can trigger condensation into the band minima and is attributed to the finite potential depth of the lattice sites (taken here to be 2 meV in calculation) which leads to deviation from the perfectly dispersionless bands predicted by the tight binding theorem [27]. The band curvature can be reduced by increasing the potential depth of the inverse Lieb lattice (see Sec. S5 in SM) which can be achieved by adjusting the system properties (e.g., exciton-photon detuning or the exciton dipole moment by appropriate choice of the semiconductor material) such that stronger pumping (bigger blueshifting reservoir) is required to achieve condensation.”

- 6) The Discussion section has been rewritten to capture our new findings in the revision of this work, to convey the message of the study clearly and its potential high impact on the scientific community. We would like to especially guide the attention of the reviewers to the new sentence:

*“We believe that our work carries significant weight in the future design and investigation of polaritonic non-Hermitian (gain and loss) lattice physics in e.g. topological lasers [I. Amelio and I. Carusotto, Phys. Rev. X **10**, 041060 (2020); P. Comaron et al., Phys. Rev. Research **2**, 022051(R) (2020)], phase transitions in many body systems [A. Zamora et al., Phys. Rev. X **7**, 041006 (2017); M. Fruchart et al., Nature **592**, 363 (2021)], non-reciprocal transport [S. Mandal et al., Phys. Rev. Lett. **125**, 123902 (2020)], and access to a multitude of gain-induced anomalies reported recently for diffractive metasurfaces [R. Kolkowski and A.F. Koenderink, IEEE **108**, 795 (2020)].”*

Reviewer 1-3: Below are some other comments:

1- Fig.4 caption needs work. Some of the referencing is inaccurate and I cannot see the "white circles in (c) and (h)" for example.

Response: We have modified Fig. 4 and corrected typos in the caption.

Reviewer 1-4: Fig.2 and Fig.3 show k vs E plots of the band structures. Why are k vs E plots not shown in Fig.4? Can we see the measured band structure?

Response: The limited spectral resolution of our experimental setup and weak polariton intensity of PL coming from surrounding energy bands belonging to the lattices in Fig. 4 did not permit us to extract dispersion data with a desirable resolution (to see all features in the spectrum) in the energy-momentum plane. Above condensation threshold we have observed bright lasing from the M points of the dispersion which correspond to the spatial condensate PL intensity pattern shown in Fig. 4f. For this reason, we did not include initially this measurement into the manuscript but presented instead experimental energy-integrated (Fig. 4g) and energy-filtered (Fig. 4l) momentum space PL plots. We stress that Fig. 4l is extracted at the condensate energy.

As requested by Reviewer 1, we have now added experimentally measured energy-momentum plane PL into Fig. S3 in the SM which complements our measurements in Fig. 4 in the main manuscript.

Reviewer 1-5: The calculated band structures in Fig.4k and Fig. S5c shows some residual band curvature. Can the authors comment on that? Is there potentially a way to improve band flatness?

Response: We thank reviewer for this question. The constructed inverse Lieb lattice is not deep enough to have complete agreement with the tight binding treatment which predicts the existence of two perfectly dispersionless bands for the P-orbitals (for the S-orbitals there is just a single dispersionless band). Instead, the finite potential depth and the Gaussian shape of the excitation beams generates a more complex potential landscape with consequent additional small curvature in the flatbands (one with negative effective mass and the other with positive). By increasing the depth of the potential minima (or conversely, the height of the potential maxima) in the lattice the flatband curvature is reduced as we demonstrate in Fig. L1 in this letter. There we show (a) the potential profile used in Bloch's theorem, (b) the calculated band structure for a 2 meV potential height, and (c) the calculated band structure for a 4 meV potential height. Going from (b) to (c) the dispersion of the flatband becomes visibly reduced. We would like to also point out the increased blueshift of the flatbands (with respect to the other bands) in Fig. L1(c). This blueshift comes from an increased energy splitting between the P_x and P_y orbitals at sublattices A and C due to the geometric shape of our lattice. That is, the inverse arrangement of Gaussians in this manner splits and shifts the P orbital levels at the sublattice sites. We have now added this explanation into the main manuscript around Fig. 4, and the data shown in Fig. L1 into a new Sec. S5 in the SM.

Figure L1: (a) Four unit cells of the pump induced potential landscape used in Bloch's analysis. (b) Calculated band structure for a peak potential height of 2 meV. (c) Calculated band structure for a peak potential height of 4 meV. The polariton mass is the same in both calculations.

Reviewer 1-6: Fig.4n and Fig.4o show condensation into two different P-bands (labelled l and m). Is this achieved by varying the laser excitation power? The lattice configuration? Is there a way to controllably decide in which mode condensation is going to occur?

Response: We thank reviewer for these questions. The different P-band states corresponding to Figs. 4n and 4o in the previous version of our manuscript are simultaneously present in the excited condensate shown in Figs. 4f and 4g in the current manuscript [previously Figs. 4h and 4i]. These states can be distinguished through spectral tomography which allows us to extract reciprocal space PL at a fixed energy (while keeping the excitation pattern and excitation density fixed). We stress that the images in Fig. 4n and Fig. 4o were independently intensity-normalized which might have given the impression that the two P-bands were similarly populated by the condensate. This is not the case and the previous Fig. 4o has now been moved from the main manuscript to the SM (see new Fig. S3d) as it was only very weakly populated and didn't carry significant weight to the message of our study (see our answer to question 1-7).

As already demonstrated in Fig. 4, control over which mode condensation occurs is possible by adjusting the geometric properties of the potential landscape. To condense the polaritons into the S-orbital lowest state we tuned the lattice constant from $D = 16 \mu\text{m}$ (corresponding to P-flatband condensation) to $D = 13 \mu\text{m}$ together with an adjustment of the excitation density per spot.

Reviewer 1-7: The authors comment on the inset of Fig.4o as being potential evidence of "a strongly interacting particle phase, inherent to a flat band state". There is not enough evidence to support this claim and the system is definitely not in the strongly interaction regime. That statement should be removed.

Response: We agree with the Reviewer that measurements presented in previous Fig. 4o were not conclusive and we have removed this statement from the manuscript and modified Fig. 4 accordingly.

Reviewer 1-8: Towards the end of the paper comes a discussion of "two faulty spots in the excitation pattern in Fig.4f". That statement is misplaced and should appear either earlier in the article (when describing the technical implementation) or in the supplemental material.

Response: We agree with this note and have removed this sentence, which contains only a minor technical detail on the experiment, from the main manuscript.

Reviewer #2 (Remarks to the Author):

Lieb lattices are well known to give rise to flatbands, especially in the case of decorated square lattices. Flatband topological states have been reported in many photonic systems, including very convenient arrangements like arrays of waveguides (ref [36]) or plasmonic scatterers (Phys. Rev. B 93, 075126 (2016)). The flatbands result from interference of the scattered coherent light with the periodicity of the scattering lattice. In this manuscript the authors offer a very small variation on the already well established literature on the subject. As a source of coherent light they use microcavity polaritons in their lasing state, and they build the decorated square lattice by locally changing the refractive index of a planar microcavity with an external laser. It is not clear, throughout the paper, what would be advantage of using polaritons operated at 4 K with respect to room temperature systems like laser arrays or waveguide arrays. On the contrary, low temperature operation looks like a stepback. While technically correct, this lack of novelty of the present work makes it more suitable for a more specialistic journal.

Response: From the comment above we identify two main reviewer's concerns:

- 1) Small variation of our work from well-established literature on the subject.
- 2) Questionable novelty and advantages of our system operating at temperature of 4 K compared to other room temperature systems.

We would like to first refer Reviewer 2 to our extensive reply to Reviewer 1 which thoroughly discusses the major changes and additional data we have introduced to the manuscript in an effort to improve the coherence and impact of our study. For brevity, we will not repeat all of our changes to our study here (although quite relevant to the broad comment of Reviewer 2) and instead focus on providing additional details directly relevant to Reviewer 2.

Starting with the first point. Indeed, the implementation of the Lieb lattices on different platforms has been discussed earlier in the literature as we highlight in Refs. [25,26,27,38-42] including realisation of the polariton Lieb lattices in lithographically engineered structures [43-47]. We have now included the suggested work of [Phys. Rev. B 93, 075126 (2016)] as

reference [42], and also a recent study [47] titled “Tunable exciton–polariton condensation in a two-dimensional Lieb lattice at room temperature”.

However, none of these platforms allow for in-situ tuning of the system between two very different regimes of condensation – from gain-localised ballistic condensates (diffractive regime $\lambda < D$) to optically trapped condensates (tight-binding regime $\lambda > D$) where λ is the mean condensate wavelength and D the lattice constant. To the best of our knowledge, neither regime has been demonstrated in two-dimensional Lieb lattices of polariton condensates. But, more crucially, the connection between the two regimes has never been explored in any polariton lattice setting to date, or other platforms to the best of our knowledge. Our system therefore offers a unique opportunity to explore non-Hermitian lattice physics at the boundary of the two regimes where the condensate mean wavelength becomes comparable to the lattice constant $\lambda \sim D$.

Indeed, it is precisely the all-optical malleability of our pump induced non-Hermitian lattice and nature of the strongly interactive polaritons which not only allows us to realise both regimes in the very same cavity system but to also accurately measure and quantify interesting physics at the boundary separating these two regimes, which has been poorly presented in literature. In the resubmitted manuscript we have enhanced our work with new experimental and theoretical data (through generalised Gross-Pitaevskii simulations) providing evidence of such an intermediate regime (see Fig. 3 in the revised manuscript). From Fig. 3l one can see the revealed transition area (white colour) between the regimes of polariton condensation on top of the pump spots (red colour) to the regime of trapped condensates between the pump spots (blue colour). Interestingly, in the intermediate regime the system demonstrates multi-mode lasing above condensation threshold, leading to unexpected polaritons distribution in real space as shown in Figs. 3b and 3c. We note that the observation of such a transition regime is practically not possible in the lithographically patterned micropillar structures, which have been the basic platform for polariton Lieb lattices so far. In order to clarify the main message and underline the novelty of our work we have incorporated these new results to the revised manuscript.

Moreover—as we now explain in the amended manuscript Introduction and to Reviewer 1—the choice of Lieb lattice is not crucial to study this transition. We could also have used e.g. honeycomb, square, triangle, etc lattices. Our choice in using a Lieb lattice is motivated by the fact that these systems are hosts to many interesting physical phenomena such as flatbands and Dirac cones, with evidence of the former phenomena presented in Fig. 4 in the manuscript. We hope that these presented results finally close the question on the novelty of our work and superficial juxtaposition with established literature.

We now address the second point of Reviewer 2 which concerns the practicality in using inorganic semiconductor polariton systems as opposed to other laser systems (e.g. room temperature polaritons or VCSEL arrays). Firstly, we hope that Reviewer 2 has not misunderstood a fundamental feature of our system. We are not changing the refractive index of the cavity which implies that the potential lattice we create is acting on the photonic

mode. Instead, as explained in the introduction, our potential lattice appears from exciting a spatial pattern of uncondensed background of almost immobile excitons and charge carriers which interact with polaritons and provide a local mean-field blueshift (i.e., a potential gradient). This means that it is the excitonic part of the exciton-polaritons which is sensing the potential landscape, not their photonic part. The obvious ingenuity in designing such a lattice is the fact that the photoexcited exciton clouds not only blueshift polaritons but provide them also with gain which puts our optical lattice into a unique position compared to other gain and loss systems like VCSEL arrays or conservative optical lattices with cold atoms.

The main advantage of our system, although operating at 4 Kelvin, is its extremely flexible, fast, non-invasive and reversible reconfiguration of the all-optical two-dimensional potential lattice of almost any arbitrary geometry. The all-optical approach that we utilised in this work allows to reshape the non-Hermitian potential landscape for the polaritons within the response time of the liquid crystal spatial light modulator used in experiment (~ 10 ms) without any destructive effects on the sample unlike lithographically written structures that would require new and costly synthesis process in order to change fine details in the lattice. Recently we have also demonstrated that by using the presented all-optical approach one can tune the couplings between the nodes of polariton condensate lattices [Alyatkin et al., Phys. Rev. Lett. **124**, 207402 (2020)]. Moreover, our polariton platform can also be scaled to larger lattices by increasing the available power to the system. We would like to stress that we are not presenting a platform for practical applications where cryogenic temperatures would be seen as a costly disadvantage, but instead a platform to explore fundamental condensed matter physics in a new sandbox environment of artificial non-Hermitian lattices.

In conclusion, we would like to thank both Reviewers for their comments and raised questions. We hope that the implemented changes in the resubmitted manuscript has improved its quality and made it appealing for publication.

Reviewers' Comments:

Reviewer #1:

Remarks to the Author:

The authors have done a tremendous work to reshape their article in order to clarify their message, and to go deeper into the analysis of their results. Their answers to the referees concerns are very satisfactory. I have no further reservations, and strongly recommend regarding publication in Nature Communications.

REVIEWERS' COMMENTS

Reviewer #1 (Remarks to the Author):

The authors have done a tremendous work to reshape their article in order to clarify their message, and to go deeper into the analysis of their results. Their answers to the referees concerns are very satisfactory. I have no further reservations, and strongly recommend regarding publication in Nature Communications.

Reply:

We thank reviewer for high evaluation of our work and recommendation to publish the revised manuscript.